# Chip Design of an All-Digital Frequency Synthesizer with Reference Spur Reduction Technique for Radar Sensing

**DOI:** 10.3390/s22072570

**Published:** 2022-03-27

**Authors:** Wen-Cheng Lai

**Affiliations:** 1Bachelor Program in Industrial Projects, National Yunlin University of Science and Technology, Douliu 640301, Taiwan; wenlai@yuntech.edu.tw or wenlai@mail.ntust.edu.tw; 2Department Electronic Engineering, National Yunlin University of Science and Technology, Douliu 640301, Taiwan

**Keywords:** all-digital, frequency synthesizer, reference spur reduction

## Abstract

5.2-GHz all-digital frequency synthesizer implemented proposed reference spur reducing with the *tsmc* 0.18 µm CMOS technology is proposed. It can be used for radar equipped applications and radar-communication control. It provides one ration frequency ranged from 4.68 GHz to 5.36 GHz for the local oscillator in RF frontend circuits. Adopting a phase detector that only delivers phase error raw data when phase error is investigated and reduces the updating frequency for DCO handling code achieves a decreased reference spur. Since an all-digital phase-locked loop is designed, the prototype not only optimized the chip dimensions, but also precludes the influence of process shrinks and has the advantage of noise immunity. The elements of novelties of this article are low phase noise and low power consumption. With 1.8 V supply voltage and locking at 5.22 GHz, measured results find that the output signal power is −8.03 dBm, the phase noise is −110.74 dBc/Hz at 1 MHz offset frequency and the power dissipation is 16.2 mW, while the die dimensions are 0.901 × 0.935 mm^2^.

## 1. Introduction

In recent years, frequency synthesizers are developing very swiftly. An all-digital frequency synthesizer has no passive elements such as resistors and capacitors in the loop filter, and it is easy to redesign with process evolutions. It operates in the digital domain, and the passive loop filter is displaced by the digital loop filter. Therefore, it eliminates the need for large area consumption and reduces the noise interference. Several studies of designing PLLs in 5.2 GHz band in recent years are found [1,2]. Literature [1] achieves good phase noise performance, but the tuning range is only 200 MHz and the settling time is long, about 51 μs. The author used a DAC to reduce the mismatch [2] for frequency synthesizer. Although the proposed technique reduces the phase noise effectively, the calibration circuits consume a large chip area and high-power consumption [3,4]. Literature [5] adopts current compensation technique and achieves better phase noise, but it needs a long settling time. Phase noise is highly important to the design of a frequency synthesizer. In order to reduce the reference spur, this proposed prototype adopts a phase detector, which delivers phase error raw data only as phase error is investigated. The phase error between the input reference clock and the feedback clock is found by the phase frequency detector (PFD), is executed by the time-to-digital converter (TDC) and is sent to the digital loop filter to control the digital-controlled oscillator (DCO) of each reference clock. When phase error is detected and revises DCO control codes at each reference clock period, the reference spur is reduced. The reference spurs are measured according to the difference in power between the carrier and the spurs at a set frequency offset given in dBc, as shown in Figure 1a [6]. A novel random clock generator is presented to perform the random selection of the phase frequency detector control for the charge pump in a locked state. The proposed all-digital frequency synthesizer is fabricated in the TSMC 0.18 μm CMOS process. For radar equipped applications and radar-communication control, measured results achieve a locked phase noise of −110.74 dBc/Hz at 1 MHz offset frequency, a power dissipation of 16.2 mW from 1.8 V souring voltage and a chip size of 0.842 mm^2^.

## 2. Circuits Architecture

The classical single-loop PLL of spur reduction [7,8,9] is published, which consists of a phase/frequency detector (PFD), a charge pump (CP), a loop filter (LF), a voltage controlled oscillator (VCO), and a frequency divider. The reference of [7] calibration method accomplishes efficient search for an optimum VCO discrete tuning curve among a group of frequency sub-bands. The agility is attributed to a proposed frequency comparison technique which is based on measuring the period difference between two signals. To maintain phase-noise optimization and loop stability over the entire output frequency range, techniques of constant loop bandwidth are proposed [8]. The proposed prototype composes of a frequency detector (FD), a phase detector (PD), a digital loop filter (DLF), a DCO, a differential to single-ended (D to S) circuit, a first-order delta-sigma modulator (DSM) and frequency dividers in the feedback path as shown in Figure 1b [10]. The proposed prototype composes of two loops, the frequency detection loop, and the phase detection loop, respectively. The former has a low resolution to perform a coarse tuning and reduce the locking time for wide tuning range operation. The latter has a high resolution to implement fine tuning. In order to upgrade the DCO resolution, a first-order DSM is used. The DCO output is connected to the current-mode logic (CML) divider to lower the frequency and connected to the D to S circuit and other dividers. When the divide ratio by a multi-modulus divider (MMD) is changed, the frequency synthesizer can obtain different frequency bands. The locked output frequency *f_DCO_* is expressed as below:(1)$$f_{DCO}=(16+N_{MMD})\times F_{REF}\times 4$$
where *N_MMD_*: Divide ratio of the programmable divider; *F_REF_*: Reference frequency.

In the frequency synthesizer, the phase error between the input reference clock and the feedback clock is detected by the phase detector (PD). The results of phase error information are delivered to the DLF each reference clock period in order to control the digital-controlled oscillator (DCO). Even though there is without phase error, the PD continues to produce signals to the DLF. Furthermore, the timing mismatch of PD, the current mismatch, and the leakage current of the DLF may downgrade the produced clock signal. These effects may result in a reference spur.

## 3. Circuits Design

The methodology of a digital-controlled oscillator (DCO) circuit in the all-digital frequency synthesizer is fundamental, and dominates the quality and usability of the signal performance. The DCO and divider with DSM produced low power consumption and phase noise. The proposed FD with DLF also consider less consumption. The reported frequency synthesizer of [9] is smaller, exhibits less phase noise, and consumes less power than prior art. In order to achieve inexpensive noise performance, prompt RF transition, and trusty lock-on to the frequency synthesizer, the significant is consider seriously to development of DCO. Since the phase noise affects directly, the out-of-band performance of a frequency synthesizer, DCO phase noise needs to be reduced. To realize the oscillator, the DCO circuit consists of PMOS cross-coupled pairs, NMOS cross-coupled pairs, a symmetric spiral inductor, and MOS varactor arrays including coarse a tuning element, a fine tune element and a ΔΣ organization as shown in Figure 2a. The differential DCO is essentially formed by two complementary NMOS and PMOS devices to generate the negative transconductance, to compensate the loss of LC tank, the advantages of more symmetric output waveform rising time and falling time and consumes less power to make it oscillate. The varactor arrays play the role of voltage-controlled capacitance to control the operation frequency of the DCO. The souring voltage of 1.8 V provides for AV_DD_, the buffer voltage, BV_DD_ and the bias V_bias_ for buffer gates and the next CML divider circuit in feedback path. In order to enhance the DCO resolution, a first-order DSM is used shown in Figure 2b in which the x[n] is k-bit input signal, and the y_1_ [n] is 1-bit overflow. The carry-out bit to control the LSB of a DCO to perform fractional control, and enhance the efficient resolution. In this design, the average period of y_1_[n] is 496.5 ps and the input signal of the DSM is 8-bit.

Key techniques present PFD to operating at frequency of 5.2-GHz with minimum dead-zone are used to meet the minimum phase offset. With this consideration shown in Figure 3a, M_5_, and M_12_ act to discharge the parasitic charges at point x whenever the point q is pulled to its logic high [11]. Therefore, parasitic-reducing M_5_, M_12_ alleviates the non-ideal output loss of the PFD. The promoted frequency detector (FD) consists of a PFD, a NOR gate, a coarse time-to-digital converter (TDC) and a decoder as depicted in Figure 3b. The PFD compares the phase error between the F_REF_ and the F_DIV_, and the NOR gate combines the phase error information up and dn, and uses TDC to quantify the information. The coarse detection has low resolution, because the FD is used for fast frequency locking. The resolution of the coarse TDC is about 340 ps. In order to cover the maximum phase error of the PFD, which is about on a half of reference clock period of 10 ns, the coarse TDC has 31 stages as shown in Figure 4a.

The proposed structure of the PD composes of a PFD, a fine TDC which has high resolution, a decoder and a register, as shown in Figure 4b. The phase difference between the F_REF_ and the F_DIV_ is detected at PFD, then the outputs of the PFD are inserted to the fine TDC. The register is used for storing the previous phase information and compares it and the current phase information. In an ADPLL, the phase error is executed by the TDC, and the phase error information is sent to the digital loop filter to control the DCO of each reference clock. When phase error is detected then updates DCO control codes at each reference clock period, the reference spur will take place. The proposed PD compares current and previous phase error information, and reduces the updating frequency of the DCO control codes and thereby reduces the reference spur. The delay chain of fine TDC is shown in Figure 5a. The leading clock is delivered to the delay chain, and the lagging clock is delivered to the D-type flip flop. The leading clock will be sampled by the lagging clock, then the phase difference of the leading and lagging clock will be digitized and quantified. The delay time of one inverter is the resolution of the fine TDC, which is about 32 ps. In order to cover the coarse TDC resolution, the stages of the fine TDC should not be less than 11, therefore, a 4-bit fine TDC is implemented in this design. Figure 5b illustrates the fine TDC structure [11]. There are two delay chains in the fine TDC; one where the leading clock is connected to the PFD output signal up, and the other one where the leading clock is connected to the PFD output signal dn. As the leading clock will be sampled by the lagging clock, a signal Sign_OUT is used to select the correct side and ensure the accuracy of the fine TDC output.

The first-order PI circuit is chosen to implement the digital loop filter. Figure 6a illustrates the s-z bilinear analog to digital domain transformation for the first-order passive RC loop filter which is a resistance and a capacitance being connected in series. A digital equivalent circuit of an analog loop filter is known as a proportional-integral (PI) controller and is chosen to implement the digital loop filter. The parameter α and β in this circuit are the proportional gain and integral gain, respectively [12]. All parameters of the digital loop filter can be defined and made a checking calculation [13]. The bilinear transform can be defined as:(2)$$s=\frac{2}{T_{s}}\;\frac{1-z^{-1}}{1+z^{-1}}$$

The DCO and the TDC are designed first, and the parameters of Δ*_TDC_*, K_DCO_, F_REF_, divide ratio, unit gain bandwidth and phase margin can be defined. Then, *I_CP_* is determined by
(3)$$I_{CP}=\frac{T_{REF}}{\Delta_{TDC}}$$

The parameters *R* and *C* can then be calculated successively by (4a) to (4d).
(4a)$$PM=\tan^{-1}(\frac{\omega_{UGBW})}{\omega_{z}})$$
(4b)$$\omega_{z}=\frac{\omega_{UGBW}}{\tan(PM)}$$
(4c)$$R=\frac{2\pi N}{I_{CP}K_{DCO}}\frac{\omega_{z}^{2}}{\sqrt{\omega_{z}^{2}+\omega_{UGBW}^{2}}}$$
(4d)$$C=\frac{\tan(PM)}{R\omega_{UGBW}}$$

Finally, instead of *R* and *C*, the proportional gain α and the integral gain β can be promoted.
(5a)$$\alpha =R-\frac{T_{s}}{2C}$$
(5b)$$\beta =\frac{T_{s}}{C}$$

The *S*-domain transfer solution of the analog loop filter is
(6a)$$H(s)=R+\frac{1}{sC}$$

Additionally, the digital loop filter transfer function in the z-domain is then expressed as
(6b)$$H(z)=\frac{G_{1}+G_{2}-G_{1}z^{-1}}{1-z^{-1}}$$

At the supply voltage of 1.8 V and reference frequency of 50 MHz, the parameters of first-order digital loop filter can then be determined successively.

The promoted divider loop consists of a CML divider, a differential to single circuit, a true single-phase clock (TSPC) divider and a MMD to achieve the divide ratio 104, as shown in Figure 6b [14]. The CML divider circuit is used to handle the higher frequency of the VCO [15,16] output signal and produces divide-by-2 signal. The circuit of differential to single-ended sends the output signal of CML to the TSPC divider, and then produces the divide-by-2. Finally, the signal sends to the MMD, and outputs the divide-by-26 feedback signal. The programmable MMD is used to produce the different frequency band of the frequency synthesizer. The programmable MMD circuit contains 2/3 divider and a logic gate as shown in Figure 7a [14]. The MMD has the ability to treat frequency division over a wide continuous range. The 16-modulus divider is used to support all integer divided ratios from 16 to 31 in this design. The divide ratio of MMD can be defined as (6). As the input frequency of MMD is 1.3 GHz, the divided ratio must be 26.
(6c)$$Divided\;Ratio=16+(2^{3}\times MC3)+(2^{2}\times MC2)+(2^{1}\times MC1)+(2^{0}\times MC0)$$

The 2/3 divider architecture consists of a two D flip flop and a traditional logic shown in Figure 7b. This design is based on the Johnson counter plus additional logic ratio selection. When control net is selected to a high level, the 2/3 divider circuit is operated in a divide-by-2 mode. Therefore, when control net is selected to a low level, the 2/3 divider circuit produces in a divide-by-3 mode.

## 4. Experiments and Discussions

The proposed 5.2 GHz all-digital frequency synthesizer is implemented in 0.18 µm CMOS process. The layout and placement are carefully report and is compact and symmetrical to avoid mismatch and shorten the output length to reduce the signal attenuation.

Figure 8a depicts the chipset microphotograph. Including wire bound pads the layout and placement dimension is 0.842 (0.901 × 0.935) mm^2^. The spectrum analyzer of Agilent E4446A and digital synthesis arbitrary function generator of Protek 9380 providing 50 MHz for reference frequency are used to perform the measurement. The proposed the DCO control divided between coarse/fine tuning, author setup parameters to quality the tradeoff between resolution and control complexity as shown in Figure 3 and Figure 4. Figure 8a of TDC is desirable to decompose DCO into multi-coarse-tune stages and fine inputs. The oscillating period key includes of delays of fine-tune-unit (FTU) and coarse-tune-unit (CTU). The TDC placement between DCO output and divider as shown in Figure 8c. At the supply voltage of 1.8 V, Figure 9a,b depicted the experimental output spectrum of free-running DCO at 4.68 GHz and synthesizer phase-locked with reference spur of 52 dB at 5.22 GHz. Base on Figure 9 measured results, the output power of free-running DCO is about −3.53 dBm, and the output power of phase-locked synthesizer is about −8.03 dBm, while the power consumption is 16.2 mW. Compared to recent literature [17], the reference spur can be reduced without any complicated circuits which need additional power or chip area. Figure 8b displays conductive measurement by probe on wafer station. At 1 MHz frequency offset, measured DCO free-running phase noise and locked synthesizer phase noise are −117.93 dBc/Hz and −110.74 dBc/Hz, respectively, as shown in Figure 10a. Measured tuning features of the DCO versus the controlled voltage varying from 0 to 1.8 V are illustrated in Figure 10b which is tunable from 4.68 to 5.36 GHz for all the possible combination of control code on FTU. Base on the wide range of frequency from 4.68 to 5.36 GHz, where it is noted that the proposed DCO possess the high linearity after multi-coarse-tune stages. To compare the proposed prototype to other works, the well-known figure-of-merit (FoM) is defined as:(7)$${\mathrm{FoM}}_{\mathrm{pn}}=PN\;(\;f_{offset}\;)-20\;\log\;(\;\frac{f_{o}}{f_{offset}}\;)+10\;\log\;(\;\frac{P_{\mathrm{DC}}}{1\;\mathrm{mW}}\;).$$
where *f_o_* is the center frequency in MHz, *f_offset_* = 1 MHz and P_DC_ is power dissipation in mW.

The appraising FoM indicates the overall quality as Formula (7). The performance is regarded to be best with a less negative or higher absolute value of the FoM. Table 1 summarizes the experimental performances of the promoted frequency synthesizer and the compares with other recently published articles. 

This work achieves the best FoM, the lowest phase noise and the best reference spur, except [20], which suffers from narrow tuning range, high power consumption and bad phase noise. Literature [17] consumes less power, wide tuning range, and optimized chip dimension, but it uses advanced 65 nm process and suffers from lower reference spur and bad phase noise. Both [17,20] achieve wide tuning range, lower consumption power, but also using advanced process and obtaining bad phase noise. Literature [21] consumes high power consumption, low phase noise, and output frequency same as proposed article. Literature [22] consumes low power consumption, low phase noise, but output frequency lower than proposed paper.

## 5. Conclusions

A 5.2 GHz all-digital frequency synthesizer is manufactured in TSMC 0.18 µm CMOS process. In order to suppress the reference spur, a phase detector which transfers phase error results only when phase error is detected has been adopted. The oscillator adopts the complementary cross-coupled LC-tank structure to reduce the power consumption. The proposed all-digital frequency synthesizer for radar equipped applications [23] and radar-communication control [24].

## Figures and Tables

**Figure 1 sensors-22-02570-f001:**
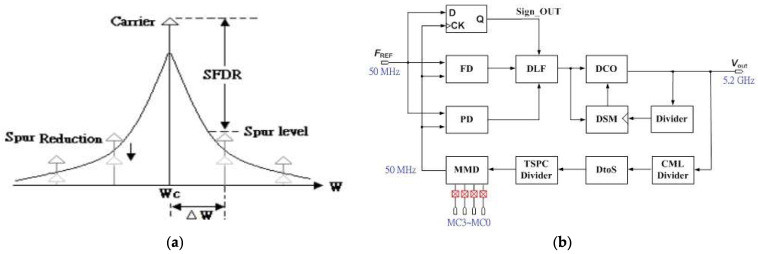
(**a**) Frequency domain representation of spurs; and (**b**) the function block of the introduced all-digital frequency synthesizer.

**Figure 2 sensors-22-02570-f002:**
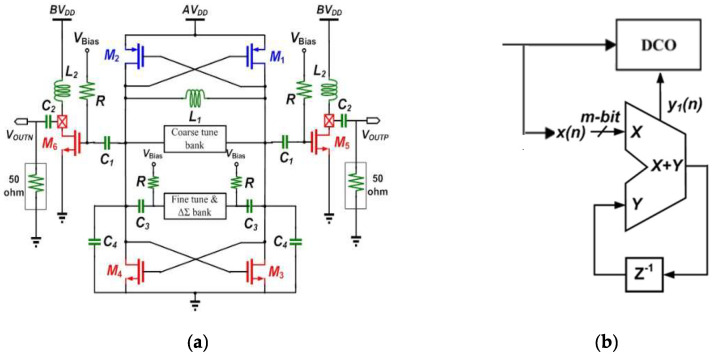
The proposed (**a**) LC-Tank VCO circuit; and (**b**) structure of the first-order delta-sigma modulator.

**Figure 3 sensors-22-02570-f003:**
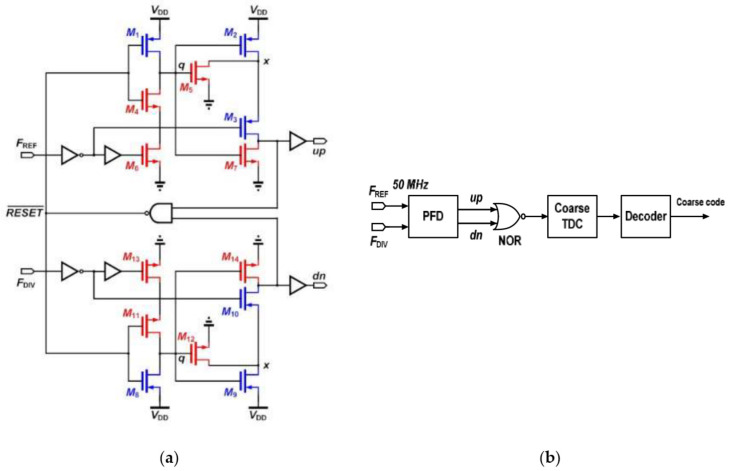
The proposed (**a**) frequency phase detector circuit; and (**b**) frequency detector structure.

**Figure 4 sensors-22-02570-f004:**
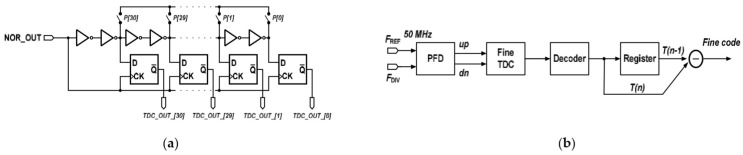
The proposed (**a**) coarse TDC structure using 31 stages; and (**b**) phase detector structure.

**Figure 5 sensors-22-02570-f005:**
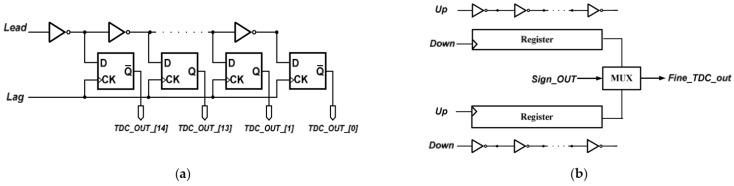
(**a**) The delay chain of the finetune TDC; and (**b**) the structure of finetune TDC.

**Figure 6 sensors-22-02570-f006:**
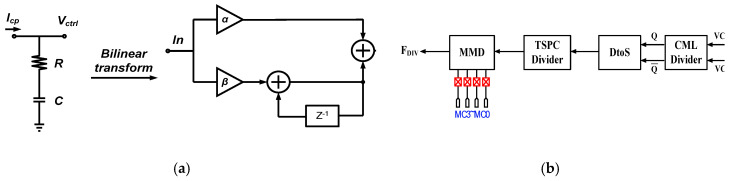
(**a**) Bilinear transformation of first-order digital loop filter for s-domain circuit and z-domain circuit.; and (**b**) the proposed programmable divider structure.

**Figure 7 sensors-22-02570-f007:**
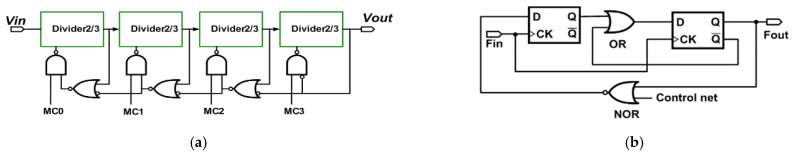
The proposed (**a**) Multiple modulus divider circuit; and (**b**) the 2/3 divider circuit.

**Figure 8 sensors-22-02570-f008:**
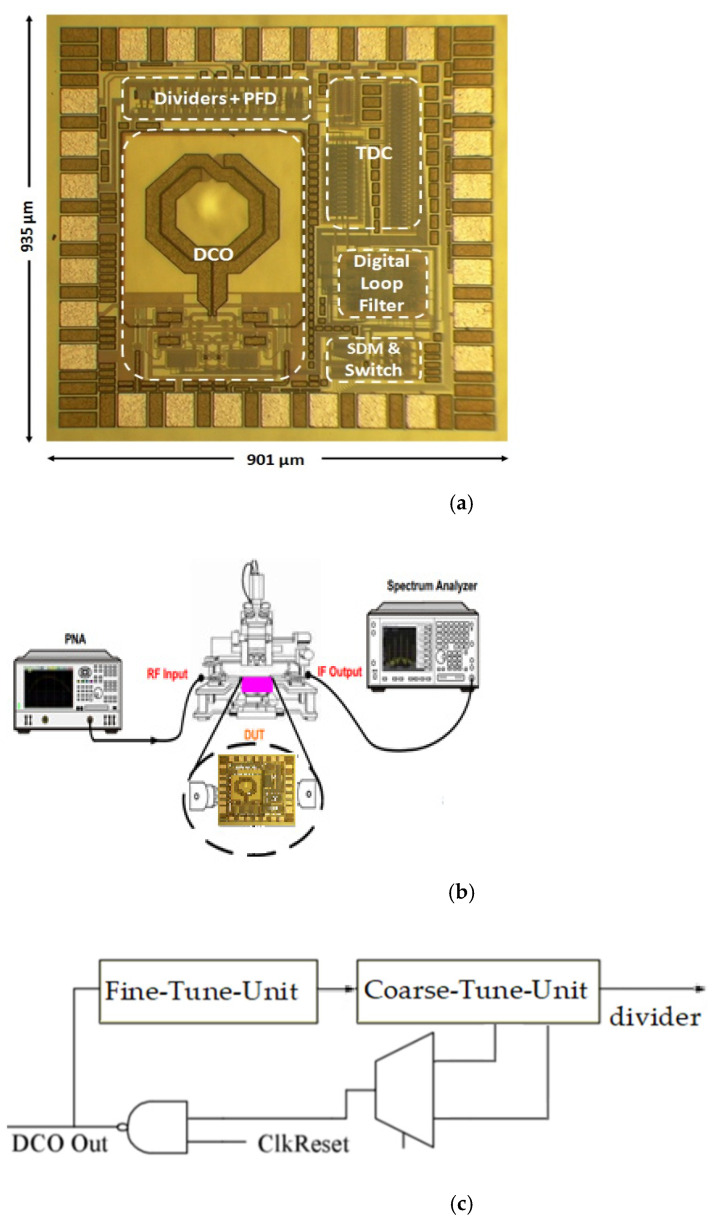
(**a**) The Die micrograph of the proposed frequency synthesizer with layout and placement square of 0.842 mm^2^. (**b**) Experimental conditions. (**c**) TDC placement drawing.

**Figure 9 sensors-22-02570-f009:**
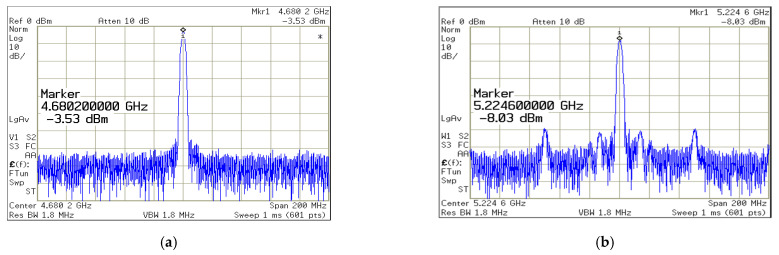
The proposed measured output spectrum (**a**) free-running DCO; and (**b**) phase-locked synthesizer with reference spur of 52 dB.

**Figure 10 sensors-22-02570-f010:**
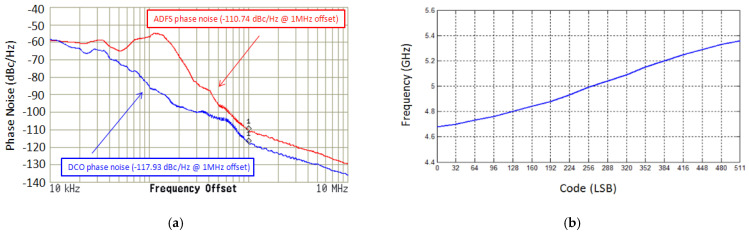
The proposed (**a**) DCO free running and synthesizer locking phase noises; and (**b**) tuning range measurement versus varying controlled−voltage.

**Table 1 sensors-22-02570-t001:** Performance comparisons with previously published papers.

	[17] (2012)	[18] (2012)	[19] (2013)	[20] (2013)	[21] 2020	[22] 2016	This
Technology (nm)	65	90	130	180	180	180	180
Synthesizer Type	ADPLL	ADPLL	ADPLL	CPPLL	ADPLL	ADPLL	ADPLL
Supply Voltage (V)	1.2	1.0/1.2	N/A	1.8	1.8	1.8	1.8
Reference Frequency (MHz)	40	60	20	20	N/A	25	52
Output Frequency (GHz)	2.7~7.3	3.57~4.3	1.9~3.1	5.18~5.32 5.74~5.82	4.83~5.47	2.23–2.47	4.68~5.36
FTR (%)	103.6	19.8	49.4	2.7 1.4	N/A	10.2	13.6
Phase Noise@1 MHz Offset (dBc/Hz)	−80	−108.25	−83.89	−102	−108.54	−111.16	−110.74
Reference Spur (dBc)	−35	N/A	N/A	−63	N/A	−60.4	−52
Loop Bandwidth (kHz)	800	700	2000	280	N/A	250	200
FOM_pn_	−82.95	−170.34	N/A	−161.81	N/A	−172.3	−172.64
Power Consumption (mW)	10	8/9.6	12 *	28.8	25.3	4.4	16.2
Chip Area (mm^2^)	0.07 **	0.34 **	0.42 **	0.764	0.788	0.72	0.842

* current consumption (mA) ** core area (mm^2^).

## Data Availability

Not applicable.

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
