# Peer review of "Chip Design of an All-Digital Frequency Synthesizer with Reference Spur Reduction Technique for Radar Sensing"

_sensors, 2022, doi:10.3390/s22072570_

Round 1

Reviewer 1 Report

All-Digital Frequency Synthesizer with reference spur reduction technique for radar sensing design is proposed with CMOS technology. The results are fair. However the manuscript still needs a number of points to be addressed.

1-The very first sentence of the abstract is not clear. Make it more meaningful.

2- The second novelty should be mentioned in the sentence (line 13) "The elements of novelties of this article are low phase noise and low phase noise."

3- Introduction section should be a detailed section encompassing all the related works. Include some more references with detailed discussions.

4- How have you come to carve this circuit design. Explain the methodology in detail.

5- Figures should be in the center of the text. The font size of the labeling in the figures should be equal to the font size of the captions.

6- The English writing of the manuscript is poor. Needs a thorough revision.

Author Response

.

  • The very first sentence of the abstract is not clear. Make it more meaningful.

ANS: The manuscript revises to be addressed as “It can be used for radar equipped applications and radar-communication control. It provides one ration frequency ranged from 4.68 GHz to 5.36 GHz for the local oscillator in RF frontend circuits.”

  • The second novelty should be mentioned in the sentence (line 13) "The elements of novelties of this article are low phase noise and low phase noise."

ANS: The elements of novelties of this article are low phase noise and low power consumption.

  • Introduction section should be a detailed section encompassing all the related works. Include some more references with detailed discussions.

ANS: The classical single-loop PLL of spur reduction [7]–[9] are published, which consists of a phase/frequency detector (PFD), a charge pump (CP), a loop filter (LF), a voltage controlled oscillator (VCO), and a frequency divider. The reference of [7] calibration method accomplishes efficient search for an optimum VCO discrete tuning curve among a group of frequency sub-bands. The agility is attributed to a proposed frequency comparison technique which is based on measuring the period difference between two signals. To maintain phase-noise optimization and loop stability over the entire output frequency range, techniques of constant loop bandwidth are proposed [8]. The reported frequency synthesizer of [9] is smaller, exhibits less phase noise, and consumes less power than prior art.

  • How have you come to carve this circuit design. Explain the methodology in detail.

ANS: The methodology of digital-controlled oscillator (DCO) circuit in the all-digital frequency synthesizer is fundamental, dominates the quality and usability of the signal performance. The DCO and divider with DSM produced low power consumption and phase noise. The proposed FD with DLF also consider less consumption.

  • Figures should be in the center of the text. The font size of the labeling in the figures should be equal to the font size of the captions.

ANS: Thanks for your comments, included in the revised manuscript.

Reviewer 2 Report

The paper presented an all-digital frequency synthesizer with reference spur reduction for radar sensing. The measurement results show the excellent performance is achieved.

However, more grammar errors can be found, so it is suggested that the english writing should be improved further.

Author Response

  1. The paper presented an all-digital frequency synthesizer with reference spur reduction for radar sensing. The measurement results show the excellent performance is achieved.

ANS: Thanks reviewer’s comments.

  1. However, more grammar errors can be found, so it is suggested that the English writing should be improved further.

ANS: Thanks reviewer’s advice, modified original grammar errors and try do best.

Reviewer 3 Report

Although I wondered at the very smart design of the all digital frequency synthesizer which opens the way to an on-chip device, at the same time I was disturbed by the quality and style of your English. In certain places, poor English style even obscures the meaning of your text, at least for those not very well acquainted with the subject. I strongly suggest a good English expert to revise the paper before publishing.

Just a couple of real typos (besides the full style):

line 14: "low phase noise" repeated

line 88: "drawn" ??

Author Response

ANS: Thanks reviewer’s advice, modified original poor English parts and try do best. And also check with good English expert later.

line 14: "low phase noise" repeated

ANS: Thanks for your comments, included in the revised manuscript.

line 88: "drawn" ??

ANS: Thanks for your comments, included in the revised manuscript.

Round 2

Reviewer 1 Report

The manuscript is revised according to the comments.